# Venous Segmental Flow Changes after Superficial Venous Intervention Demonstrating by Quantitative Phase-Contrast Magnetic Resonance Analysis: Preliminary Data from a Longitudinal Cohort Study

**DOI:** 10.3390/jpm12061000

**Published:** 2022-06-19

**Authors:** Chien-Wei Chen, Yuan-Hsi Tseng, Chih-Chen Kao, Yeh Giin Ngo, Chung-Yuan Lee, Teng-Yao Yang, Yu-Hui Lin, Yao-Kuang Huang

**Affiliations:** 1Department of Diagnostic Radiology, Chia Yi Chang Gung Memorial Hospital, Chiayi 61363, Taiwan; chienwei33@gmail.com; 2College of Medicine, Chang Gung University, Taoyuan 33302, Taiwan; cckaomd@gmail.com (C.-C.K.); genius17ngo@gmail.com (Y.G.N.); yuan8810@cgmh.org.tw (C.-Y.L.); 2859@adm.cgmh.org.tw (T.-Y.Y.); 3Department of Obstetrics and Gynecology, Chia Yi Chang Gung Memorial Hospital, Chiayi 61363, Taiwan; 8802003@cgmh.org.tw; 4Division of Thoracic and Cardiovascular Surgery, Chia Yi Chang Gung Memorial Hospital, Chiayi 61363, Taiwan; 5Division of Thoracic and Cardiovascular Surgery, Chia Yi Hospital, MOHW, Chiayi 61363, Taiwan; 6Cardiology, Chia Yi Chang Gung Memorial Hospital, Chiayi 61363, Taiwan; vw200162@gmail.com

**Keywords:** magnetic resonance imaging, non contrast, phase contrast, hemodynamics, objective, postop

## Abstract

The effects of superficial venous intervention on hemodynamics can be quantified using two-dimensional phase-contrast magnetic resonance imaging (2D PC-MRI). Twelve patients received pre- and postintervention 2D PC-MRI analysis using quantitative hemodynamic parameters. Fifteen healthy volunteers served as controls. The 2D PC-MRI results of the target limbs (limbs scheduled for intervention for venous reflux) differed from those of the controls in terms of stroke volume (SV), forward flow volume (FFV), absolute stroke volume (ASV), and mean flux (MF) in all venous segments. The velocity time integral (VTI) and mean velocity (MV) of the popliteal vein (PV) segments were similar between the target limbs and controls preoperatively. After intervention, the target limbs exhibited an increase in VTI and MV in the femoral vein (FV) and PV segments. We compared the target and nontreated limbs of the individual patients preoperatively and postoperatively to minimalize individual bias. All QFlow parameter ratios in the FV segment increased after venous intervention (VTI, *p* = 0.025; MV, *p* = 0.024). In the PV segment, FFV and ASV increased significantly (*p* = 0.035 and 0.024, respectively). After interventions, the volume (FFV and ASV) of the PV segment and the efficiency (VTI and MV) of the FV segment significantly increased.

## 1. Introduction

Few objective diagnostic modalities for venous disorders are currently available. The magnetic resonance angiography (MRA) using gated three-dimensional (3D) turbo spin-echo short tau inversion recovery (TSE-STIR) sequence has been used to diagnose leg venous diseases since 2017 and was demonstrated to be efficacious in combination with ultrasonography (US) in a preliminary report [1,2,3]. The morphology of the entire venous anatomy of lower extremities, particularly that of the low-flow superficial venous system and pelvic collaterals in various diseases, can be observed through 3D imaging without a contrast medium or radiation [4,5]. Gated 3D-TSE-STIR MRA combing with hemodynamic analysis, by using two-dimensional (2D) phase-contrast (PC) magnetic resonance imaging (MRI) scanning, has been the standard preoperative evaluation method for the venous diseases of the legs at our institution [4,5,6,7,8,9,10]. In this study, we explored the role of 2D PC-MRI quantitative analysis plus gated 3D-TSE-STIR MRA in managing leg venous diseases and investigated the hemodynamic effects of superficial venous interventions involving the truncal ablation of the diseased great saphenous vein.

## 2. Materials and Methods

### 2.1. Patients

The Institutional Review Board of Chang Gung Memorial Hospital approved this study (numbers 201802137B0, 201901058B0, and 202100938B0). This study enrolled consecutive patients evaluated for the venous pathology of their lower extremities through non-contrast MRI, including gated 3D-TSE-STIR MRA and 2D PC-MRI scanning, at a tertiary hospital between April 2017 and October 2021. We prospectively collected and retrospectively analyzed their clinical data. All patients were suspected to have venous pathology in their lower extremities. Initially, 271 patients (including 15 healthy volunteers) were evaluated, and 21 patients were excluded. The exclusion criteria were being pregnant, having restless legs, having extreme arrhythmia, being morbidly obese, and having ferromagnetic devices incompatible with MRI. In the remaining 250 patients, anatomic evaluation was performed using gated 3D-TSE-STIR MRA and hemodynamic evaluation was performed using 2D PC-MRI quantitative analysis. On the basis of their symptoms and MRI schedule indication, the 250 patients were divided into five groups, namely superficial venous varicose with venous reflux, stasis leg ulcers, swollen legs favoring venous occlusion, other presentation, and healthy volunteers (Figure 1). According to the 2D PC-MRI analysis, 116 patients had venous reflux in their legs. A total of 60 patients received superficial venous interventions and 12 underwent postintervention gated 3D-TSE-STIR MRA plus 2D PC-MRI analysis.

### 2.2. MRI Acquisition and Phase-Contrast Hemodynamic Analysis

MRI was performed using a 1.5 T MRI scanner (Philips Ingenia, Philips Healthcare, Best, The Netherlands). During the examination, the patient must be equipped with an electrocardiography detector and a respiration detector. Technicians use the gating method to detect the heart rhythm and then use the heart synchronization method to obtain MRI images in the supine position (Figure 2). Gated 3D-TSE-STIR MRA imaging was performed using the following parameters: repetition time, 1 beat; echo time, 85 ms; inversion recovery delay time, 160 ms; voxel size, 1.7 mm × 1.7 mm × 4 mm; and field of view, 360 mm × 320 mm. STIR provides extra background suppression because of the additional suppression of the fat and bones. When systolic triggering was applied, the arteries appeared black. The imaging process yielded a 3D data set of the venous system. A trans-axial PC-MRI scan was routinely performed at the aortic bifurcation level to determine the triggering time for imaging acquisition. Hemodynamic analysis using 2D PC-MRI scanning was performed by the following parameters: repetition time, shortest; echo time, shortest; flip angle, 20 degrees; 25 im-ages/period; acquisition matrix, 324 × 324; slice thickness, 5 mm; pixel size, 0.33 mm × 0.33 mm; and velocity encoding, 80 cm/s. Trans-axial scanning of 2D PC-MRI was performed at the venous segments, including the inferior vena cava, external iliac veins (EIVs), femoral vein (FV), popliteal vein (PV), and great saphenous veins (GSV). By drawing the region of interest (ROI) on the vascular lumens (covering the whole lumen), the hemodynamic parameters will be generated by measuring the phase-shifting information of voxels within the ROI. All of the hemodynamic parameters are shown as follows: stroke volume (SV: the net volume of blood that passes through the contour of ROI during one heartbeat), forward flow volume (FFV: the volume of blood that passes through the contour of ROI in the positive direction (toward head direction) during one heartbeat), absolute stroke volume (ASV: the absolute value of forwarding flow volume plus the absolute value of backward flow volume), mean flux (MF: stroke amount × heartbeat/60 (one-heartbeat)), velocity time integral (VTI, also known as stroke distance: the net distance that blood proceeds in the vessel during one heartbeat), and mean velocity (MV: stroke distance × heartbeat/60 (one-heartbeat)). The whole process of non-contrast MRI requires 25 min for imaging acquisition, reconstructing imaging, and hemodynamic analysis.

### 2.3. Statistical Analysis

Continuous variables (age and hemodynamic parameters) were analyzed using an unpaired two-tailed Student’s t test or one-way analysis of variance, and discrete variables (sex, substance usage, comorbidities, and intervention history) were compared using a two-tailed Fisher’s exact test. All statistical analyses were performed using Stata statistics and data analysis (version 8.0; Stata Corporation, College Station, TX, USA). The results are presented as means and standard deviations. Statistical significance was defined as *p* < 0.05.

## 3. Results

A total of 60 patients received superficial venous interventions and 12 received postintervention gated 3D-TSE-STIR MRA plus 2D PC-MRI analysis. Table 1 lists the sex, age, comorbidities, morbid leg, Clinical–Etiology–Anatomy–Pathophysiology classification, preop venous clinical severity score (VCSS), wound status, and interventions of the 12 patients. Four patients presented at the clinic with static leg ulcers, and their VCSS ranged from 7 to 20. They all had primary reflux limited to the superficial leg veins and received unilateral truncal ablation by either radiofrequency, laser (Wolf ARC catheters), or cyanoacrylate ablation (VenaSeal). The 12 patients underwent 2D PC-MRI analysis pre-operatively and then received a second 2D PC-MRI analysis from the third to sixth months after the superficial venous intervention. The 15 healthy volunteers received 2D PC-MRI analysis as the control.

### 3.1. Comparison between Target Limbs and Health Control by 2D PC-MRI Quantitative Analysis: Preintervention and Postintervention

The target limbs with venous reflux exhibited different 2D PC-MRI patterns from those of the healthy controls in terms of SV, forward flow volume (FFV), ASV, and MF in all venous segments (Table 2).

The VTI and MV of the PV segment were similar between the target limbs and those of the healthy controls preoperatively. After intervention, the treated limbs exhibited increased VTI and MV in the FV/PV segment, and the difference between the treated limbs and healthy controls was significant (Table 3).

### 3.2. Ratio of Target Limb to Nontreated Limb in Each Individual: Effects of Venous Intervention

To minimalize individual bias, we compared the targeted and nontreated limbs of the individual patients preoperatively and postoperatively (Table 4). The EIV segment differed in terms of all 2D PC-MRI parameter ratios (targeted limbs to nontreated limbs) after venous intervention. All 2D PC-MRI parameter ratios (SV, FFV, ASV, MF, VTI, and MV) in the FV segment increased after venous intervention, and VTI (*p* = 0.025) and MV (*p* = 0.024) increased significantly. In the PV segment, the SV ratio increased (*p* = 0.058) and FFV and ASV increased significantly (*p* = 0.035 and 0.024, respectively). The MF, VTI, and MV ratio in the PV segment increased after intervention (*p* = 0.059, 0.08, and 0.082, respectively). All 2D PC-MRI parameter ratios decreased significantly in the GSV segments, indicating the effectiveness of ablation.

## 4. Discussion

Arterial disorders, such as occlusion and aneurysmal changes, are easily diagnosed and treated. Unlike arterial lesions, venous disorders are complex, have slow therapeutic responses, and often present as a sentinel of neoplasm, hypercoagulation, infection, and autoimmune disorders. Superficial venous reflux, mostly involving GSV reflux, is the most frequent form of venous insufficiency in symptomatic patients and is often responsible for varicose veins and stasis ulcers of the lower extremities. The introduction of endovenous techniques almost 20 years ago changed the treatment of varicose veins [11,12,13]. Minimally invasive technology, such as segmental radiofrequency, hemoglobin-specific lasers, water-specific lasers, endovenous steam, and mechanochemical and cyanoacrylate ablation, has been proven to be safe and is becoming more popular than the standard surgical method (high ligation and stripping) [11,14,15,16,17,18,19].

By contrast, with advancements in the minimally invasive techniques of venous surgery, little progress has been made in terms of diagnostic methods. In addition, no objective tool for the assessment of the post-venous intervention of the lower limbs is currently available [20]. US is a frequently used tool for the initial diagnosis of venous diseases of the legs and to assess the results of venous intervention. However, it is operator dependent and provides minimal information regarding the pelvic area. Conventional venography is the first objective tool for the detection of venous lesions; however, its invasiveness prevents its widespread use. Computed tomography (CT) venography is useful for the exclusion of pulmonary embolism in patients with signs of thrombosis in the legs and abdominal lesions, but it cannot replace US for the detection of deep venous thrombosis in the legs [21,22]. Injected contrast media are often unevenly distributed through the venous system during CT venogram acquisition. In CT venography, contrast media are injected into morbid limbs to optimize venous image quality, which may harm morbid legs. 

Numerous MRA techniques are used to reconstruct vascular structures. Developed in 1998, time-of-flight MRA was the earliest technique for evaluating arterial pathology [23]. However, it requires considerable time to produce an entire image of the lower extremity and has thus become less clinically applicable [24,25,26]. MRA with gadolinium-based contrast media is a rapid method of imaging the lower extremities [27,28]. However, gadolinium-based contrast media might cause nephrogenic sclerosing fibrosis [29,30]. The gated 3D-TSE MRA technique, which is used to identify differences in the vascular signal intensity during the cardiac cycle through image subtraction, was first applied in 1985 [31]. Gated 3D-TSE MRA has frequently been used to diagnose cranial neurologic diseases and most arterial diseases; however, it has few known applications for venous pathology and particularly for the lower extremities [32,33,34,35]. We demonstrated that gated 3D-TSE-STIR MRA, plus the STIR sequence to provide additional background subtraction of bone and soft tissue, offers several advantages in evaluating lower extremity venous pathology. First, it provides an objective 3D model of the venous morphology, including the pelvis and caves. Second, it does not require venipuncture, radiation, or contrast medium injection. Third, as this study demonstrated, hemodynamics can be recorded and evaluated when required.

We evaluated legs with venous reflux scheduled for GSV truncal ablation (target legs) through gated 3D-TSE-STIR MRA plus 2D PC-MRI scanning and compared the resulting data with those of healthy controls. The target legs exhibited higher SV, FFV, ASV, and MF values in all venous segments, which is consistent with our previous findings (Table 2). VTI and MV were higher in the GSV segments of the target limbs (*p* = 0.055 and 0.044, respectively) but similar in the PV segments (*p* = 0.223 and 0.201, respectively). After truncal ablation, the VTI and MV in the PV segment of target limbs became higher than those of the controls (*p* = 0.001 and 0.001, respectively; Table 3). These results suggest that increasing the volume may lead to more efficient and faster transport though the PV after ablation of the diseased GSV. We compared the target and nontreated limbs of the individual patients preoperatively and postoperatively to minimalize individual bias (Table 4). We observed that the FFV and ASV increased significantly in the PV segment after intervention (*p* = 0.035 and 0.024, respectively), with similar FFV and ASV values in the EIV segments. VTI and MV increased in the FV segment more than they did in the PV segment (*p* = 0.025 and 0.024 vs. *p* = 0.08 and 0.082, respectively). These results indicate faster and more efficient volume transport though the FV segments after the intervention. A less significant increase in VTI and MV in the PV segment suggests higher complexity in the pathophysiology of the interaction between the perforator veins and lessor saphenous systems.

We used gated 3D-TSE-STIR MRA and 2D PC-MRI quantitative analysis to manage patients receiving scheduled truncal ablation for superficial venous reflux. We compared 2D PC-MRI patterns among the target limbs, nontarget limbs, and the limbs of the healthy controls after intervention. The 2D PC-MRI parameters were higher in the target limbs with reflux than in those of the healthy controls. The volume (FFV and ASV) of the PV segment and the efficiency (VTI and MV) of the FV segment increased significantly after superficial venous reflux intervention. 

There are several technical issues of 2D PC-MRI to be discussed. First, this study proposes an application of 2D PC-MRI in lower extremity venous disease, which is still poorly reported and not widely understood. This study performed hemodynamic analysis using 2D PC-MRI scanning on axial planes of the venous segments. By drawing the region of interest (ROI) on the vascular lumens (covering the whole lumen), the hemodynamic parameters will be generated by measuring the phase-shifting information of voxels within the ROI. When using quantitative 2D PC-MRI analysis for evaluating the superficial venous system, stroke volume reflects the net volume of blood that passes through the contour of ROI during one heartbeat. As well as that, forward flow volume reflects the volume of blood that passes through the contour of ROI only in the positive direction (toward the head direction) during one heartbeat. Second, venous flow is phasic, frequently a combination of cardiac and respiratory cycles. Making acquisitions during a cardiac cycle may give a random value. To determine the correct time during the repeat processes for imaging acquisition, technicians use the gating method to detect the heart and respiratory cycles. Thus, MRI data from the consistent phase can be acquired. Another issue is that venous flow is sensitive to muscle contractions, especially in the popliteal vein. In our experience, in most cases, the lower extremity venous system can be imaged clearly by MRI in the supine position without being affected by muscle contractions. Our MRI protocol adopts the supine position because this position is often used as the standard position for imaging acquisitions. In our experience, the entire venous system of the lower extremities can be displayed clearly without being affected by muscle contractions in most cases. Third, beyond the presented result, we performed the statistical analysis to compare the difference between the target limb and the contralateral leg during the initial investigation. However, our data showed no significant difference between the target limb and the contralateral leg. We suspect it may be due to some subjects having venous reflux in both legs. Thus, we compared the differences between the diseased leg and the healthy controls and between the diseased leg and the treated leg.

### Study Limitations

The major limitations of this study are its nonrandomized design and small sample size. Although we performed the MRI on 271 subjects, only 12 subjects completed the postintervention MRI. Larger sample sizes, long follow-up intervals for clinical correlations, and a randomized study design may provide more solid results.

## 5. Conclusions

Gated 3D-TSE-STIR MRA plus 2D PC-MRI quantitative analysis is helpful in venous disorders and has a novel application in postop hemodynamic records after venous intervention. After the truncal ablation of the diseased GSV, the volume (FFV and ASV) of the PV segment and the efficiency (VTI and MV) of the FV segment increased significantly.

## 6. Patents

Invention I740594 in the Taiwan Intellectual property Office, from 21 September 2021 to 3 August 2040.

## Figures and Tables

**Figure 1 jpm-12-01000-f001:**
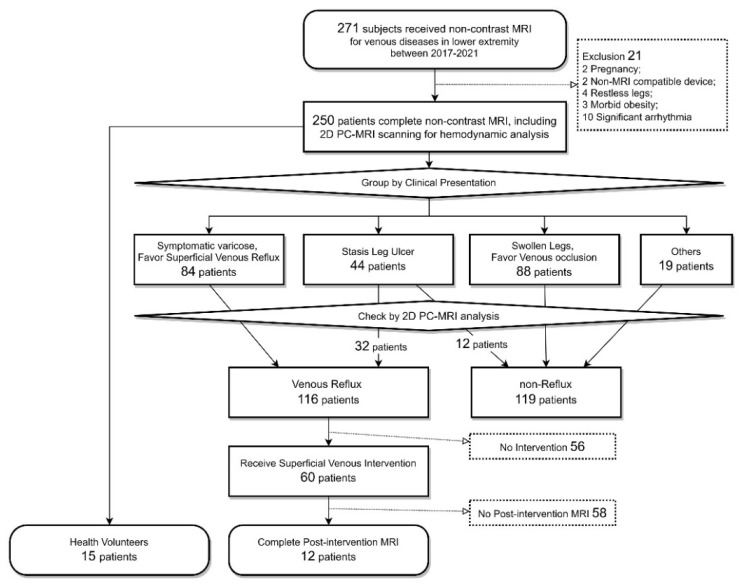
Cohorts of the study.

**Figure 2 jpm-12-01000-f002:**
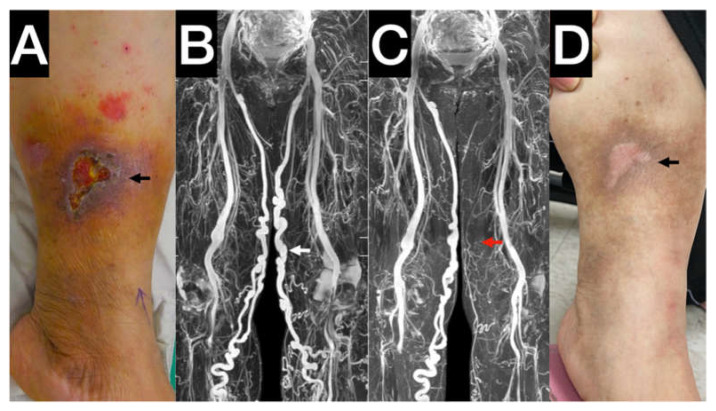
Typical stasis leg ulcer with venous reflux in the gaiter area (Also seen in Appendix A). (**A**) Wet and nonhealing stasis leg ulcer. Black Arrow: unhealed venous ulcer with cellulitis. (**B**) Preintervention: The gated 3D-TSE-STIR MRA demonstrates the venous anatomy both leg veins. The white arrow indicates the diseased great saphenous vein of the wounded leg. (**C**) Postintervention: gated 3D-TSE-STIR MRA. The red arrow indicates the diseased great saphenous vein occluded. (**D**) Stasis leg ulcer healed after truncal ablation. Gated 3D-TSE-STIR MRA for venous systems. Black arrow: healed ulcers.

**Table 1 jpm-12-01000-t001:** Demographic data of the 12 patients have TRANCE MR venous mapping pre and post interventions.

No	Age	Sex	Comorbidities	Target Legs	Main Symptoms	C in CEAP	E in CEAP	A in CEAP	P in CEAP	Wound Location	VCSS	Surgical Intervention
1	59	F	DM	Left	Wound	C6	Ep	GSVa, GSVb, SSV, CPV	Pr	gaiter area	19	A.R.C catheter ablation
2	50	M	Nil	Right	Wound	C6	Ep	GSVa, GSVb	Pr	gaiter area	13	A.R.C catheter ablation
3	48	F	Nil	Left	pain, swelling	C5	Ep	GSVa, GSVb	Pr	nil	10	A.R.C catheter ablation
4	61	F	Cervix cancer	Left	pain, claudication	C5	Ep	GSVa, GSVb, LSV	Pr	nil	9	Atoven cathter ablation, sclerotherapy
5	55	F	Cholesteremia, Hepatitis B	Left	pain, swelling, mild skin hyperpigmentation	C5	Ep	GSVa, GSVb	Pr	nil	7	A.R.C catheter ablation
6	56	F	Hypertension, tongue SCC	Left	swelling	C5	Ep	GSV, LSV	Pr	nil	10	A.R.C catheter ablation
7	65	F	Nil	Left	cramping pain, heat	C5	Ep	GSVa, GSVa, GSVbLSV	Pr	nil	7	Venaseal
8	59	F	Nil	Right	red swelling, itchy patches	C5	Ep	GSVa, GSVb	Pr	nil	7	Venaseal
9	43	M	DM, Hypertension	Right	Wound, pain	C6	Ep	GSVa, GSVb, LSV	Pr	gaiter area	17	A.R.C catheter ablation
10	72	F	DM, Hypertension	Left	swelling	C5	Ep	GSVa, GSVb, LSV	Pr	nil	10	A.R.C catheter ablation
11	59	M	Nil	Left	cramping	C5	Ep	GSVa, GSVb	Pr	nil	8	Atoven catheter ablation, sclerotherapy
12	65	F	Renal insufficiency, DM, Hypertension, Hepatitis C	Right	Wound, pain	C6	Ep	GSV, SSV	Pr	gaiter	20	A.R.C catheter ablation

AASV: anterior accessory saphenous vein; CEAP: Clinical-Etiology-Anatomy-Pathophysiology; CHF: congestive heart failure; CPV: calf perforator vein; CVA: cerebral vascular accident; DM: diabetes mellitus; F: female; HTN: hypertension; GSVa: great saphenous vein above knee; GSVb: great saphenous vein below knee; M: male; MR: mitral regurgitation; SSV: short saphenous vein.

**Table 2 jpm-12-01000-t002:** Hemodynamics parameter between target limb (before intervention) and healthy controls.

QFlow	Segments	Target Limb before Interventions		Heathy Controls		*p* Value
		Mean	SD	Mean	SD	
Stroke volume (SV), mL						
	EIV	5.898	2.200	3.751	1.386	< 0.001
	FV	1.728	0.801	1.124	0.511	0.004
	PV	1.095	0.446	0.602	0.338	< 0.001
	GSV	0.936	0.518	0.310	0.219	0.001
Forward flow volume (FFV), mL						
	EIV	5.932	2.172	3.872	1.506	0.001
	FV	1.731	0.797	1.139	0.505	0.004
	PV	1.102	0.441	0.614	0.331	< 0.001
	GSV	0.941	0.513	0.321	0.209	0.001
Absolute stroke volume (ASV), mL						
	EIV	5.956	2.154	4.022	1.602	0.002
	FV	1.734	0.792	1.155	0.502	0.005
	PV	1.108	0.437	0.625	0.325	< 0.001
	GSV	0.947	0.507	0.334	0.199	0.001
Mean flux (MF), mL/s						
	EIV	6.641	2.958	4.144	1.752	0.015
	FV	1.979	1.019	1.224	0.604	0.003
	PV	1.223	0.544	0.661	0.401	< 0.001
	GSV	1.037	0.524	0.336	0.240	0.001
Velocity time integral (VTI), cm						
	EIV	6.653	2.114	3.856	1.764	< 0.001
	FV	4.190	2.219	3.004	1.677	0.057
	PV	1.464	0.948	1.154	0.704	0.233
	GSV	2.768	1.853	1.570	1.256	0.055
Mean velocity (MV), cm/s						
	EIV	7.288	2.123	12.630	49.991	0.715
	FV	4.694	2.574	3.251	1.859	0.040
	PV	1.611	1.094	1.241	0.765	0.201
	GSV	3.088	2.088	1.669	1.272	0.044

EIV: external iliac vein; GSV: great saphenous vein; FV: femoral vein; PV: popliteal vein.

**Table 3 jpm-12-01000-t003:** Hemodynamics parameter between target limb (after intervention) and healthy controls.

QFlow	Segments	Target Limb after Interventions		Heathy Cotrols		*p* Value
		Mean	SD	Mean	SD	
Stroke volume (SV), mL						
	EIV	6.334	2.093	3.751	1.386	< 0.001
	FV	2.671	1.560	1.124	0.511	0.006
	PV	1.750	0.928	0.602	0.338	0.001
	GSV	0.054	0.070	0.310	0.219	< 0.001
Forward flow volume (FFV), mL						
	EIV	6.413	2.025	3.872	1.506	< 0.001
	FV	2.843	1.360	1.139	0.505	0.001
	PV	1.750	0.928	0.614	0.331	0.001
	GSV	0.096	0.046	0.321	0.209	< 0.001
Absolute stroke volume (ASV), mL						
	EIV	6.490	1.979	4.022	1.602	< 0.001
	FV	2.849	1.353	1.155	0.502	0.001
	PV	1.750	0.928	0.625	0.325	0.001
	GSV	0.186	0.067	0.334	0.199	0.001
Mean flux (MF), mL/s						
	EIV	7.276	2.959	4.144	1.752	0.004
	FV	3.245	1.728	1.224	0.604	0.002
	PV	1.989	1.125	0.661	0.401	0.002
	GSV	0.049	0.059	0.336	0.240	< 0.001
Velocity time integral (VTI), cm						
	EIV	5.673	1.806	3.856	1.764	0.004
	FV	6.225	4.332	3.004	1.677	0.027
	PV	1.959	0.580	1.154	0.704	0.001
	GSV	0.063	0.439	1.570	1.256	< 0.001
Mean velocity (MV), cm/s						
	EIV	6.373	2.224	12.630	49.991	0.669
	FV	7.160	5.292	3.251	1.859	0.028
	PV	2.188	0.747	1.241	0.765	0.001
	GSV	0.081	0.386	1.669	1.272	< 0.001

EIV: external iliac vein; GSV: great saphenous vein; FV: femoral vein; PV: popliteal vein.

**Table 4 jpm-12-01000-t004:** Parameter ratio of target to non-treated limb: before and after intervention.

QFlow	Segments	Ratio before Interventions		Ratio after Interventions		*p* Value
		Mean	SD	Mean	SD	
Stroke volume (SV), mL						
	EIV	0.984	0.330	0.912	0.298	0.437
	FV	1.124	0.342	1.522	1.061	0.164
	PV	1.115	0.589	1.799	0.955	0.058
	GSV	2.829	2.393	0.362	0.861	0.007
Forward flow volume (FFV), mL						
	EIV	0.918	0.344	0.897	0.314	0.835
	FV	0.966	0.410	1.593	1.081	0.125
	PV	1.065	0.482	1.793	0.956	0.035
	GSV	2.515	2.080	0.336	0.416	0.002
Absolute stroke volume (ASV), mL						
	EIV	0.958	0.334	0.888	0.326	0.456
	FV	1.103	0.337	1.652	1.035	0.056
	PV	1.026	0.409	1.788	0.957	0.024
	GSV	2.029	1.834	0.499	0.562	0.014
Mean flux (MF), mL/s						
	EIV	0.984	0.329	0.913	0.301	0.446
	FV	1.121	0.346	1.631	0.976	0.053
	PV	1.117	0.588	1.791	0.943	0.059
	GSV	2.941	2.599	0.347	0.807	0.008
Velocity time integral (VTI), cm						
	EIV	1.041	0.401	1.054	0.425	0.938
	FV	1.060	0.298	1.806	1.107	0.025
	PV	1.150	0.800	1.793	0.826	0.080
	GSV	1.482	1.033	0.421	0.911	0.027
Mean velocity (MV), cm/s						
	EIV	1.042	0.401	1.053	0.424	0.945
	FV	1.059	0.298	1.799	1.091	0.024
	PV	1.150	0.807	1.788	0.816	0.082
	GSV	1.483	1.039	0.423	0.916	0.027

EIV: external iliac vein; GSV: great saphenous vein; FV: femoral vein; PV: popliteal vein.

## Data Availability

The data presented in this study are available on request from the corresponding author. The data are not publicly available due to ethical restrictions.

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
