# Peer review of "Venous Segmental Flow Changes after Superficial Venous Intervention Demonstrating by Quantitative Phase-Contrast Magnetic Resonance Analysis: Preliminary Data from a Longitudinal Cohort Study"

_jpm, 2022, doi:10.3390/jpm12061000_

Round 1
Reviewer 1 Report
1.**The manuscript is lacking definitions of hemodynamic parameters. All of them should be clearly defined: SV, FFV, ASV, MF, Stroke distance (please use an abbreviation other than SD to not be mistaken for Standard Deviation), MV. For example, how was the “stroke volume” defined in veins. Venous flow is phasic, frequently a combination of cardiac and respiratory cycles, or just respiratory cycle. Making acquisitions during a cardiac cycle may give a random value. In addition, venous flow, especially in the popliteal vein is extremely sensitive to muscle contractions. How were these factors addressed?
2.** QFlow tables have no units. The units of measurements (especially for flow volume) should be listed and clearly defined.
3.** The rational for indexing QFlow parameters to the contralateral leg should be discussed. How the cases with bilateral disease (figure 2) were handled in the analysis?
Author Response
Comment 1.
The manuscript is lacking definitions of hemodynamic parameters. All of them should be clearly defined: SV, FFV, ASV, MF, Stroke distance (please use an abbreviation other than SD to not be mistaken for Standard Deviation), MV. For example, how was the “stroke volume” defined in veins. Venous flow is phasic, frequently a combination of cardiac and respiratory cycles, or just respiratory cycle. Making acquisitions during a cardiac cycle may give a random value. In addition, venous flow, especially in the popliteal vein is extremely sensitive to muscle contractions. How were these factors addressed?
Response to comment 1:
(i) I had added the definitions of hemodynamic parameters.
(ii) The term “Stroke Distance” was replaced with the term “Velocity Time Integral (VTI), which was the synonym of Stroke Distance".
(iii) Venous flow is phasic, frequently a combination of cardiac and respiratory cycles. Making acquisitions during a cardiac cycle may give a random value. To determine the correct time during the repeat processes for imaging acquisition, technicians use the gating method to detect the heart and respiratory cycles. Thus, MRI data from the consistent phase can be acquired. Another issue is venous flow is sensitive to muscle contractions, especially in the popliteal vein. In our experience, in most cases, the lower extremity venous system can be imaged clearly by MRI in the supine position without being affected by muscle contractions. Our MRI protocol adopts the supine position because this position is often used as the standard position for imaging acquisitions. In our experience, the entire venous system of the lower extremities can be displayed clearly without being affected by muscle contractions in most cases. We added description in the Discussion section.
Comment 2. QFlow tables have no units. The units of measurements (especially for flow volume) should be listed and clearly defined.
Response to comment 2: The units of the hemodynamic parameters had been added in table 2-4.
Comment 3. The rational for indexing QFlow parameters to the contralateral leg should be discussed. How the cases with bilateral disease (figure 2) were handled in the analysis?
Response to comment 3: Beyond the presented result, we performed the statistical analysis to compare the difference between the target limb and the contralateral leg during the initial investigation. However, our data showed no significant difference between the target limb and the contralateral leg. We suspect it may be due to some subjects having venous reflux in both legs. Thus, we compared the differences between the diseased leg and the healthy controls and between the diseased leg and the treated leg. We added description in detail at the Discussion section.
Thank you for the informative and careful reviewing our manuscript. We learn lot during the revision this article.
Reviewer 2 Report
I read with great interest the paper. The study aimed to determine if the effects of superficial venous intervention on hemo-dynamics could be quantified using QFlow sequence of MRI. They confirm on 12 patients differences in measured parameters between patients and healthy controls as well as in patients before and after interventions.
The topic is very interesting, the use of (primarily) cardiac sequence for venous system of lower extremities rather innovative. Using this approach offers new insight into venous system during interventions. The paper is clearly written and the results are properly discussed. The paper confirms that CMR can provide clinically relevant information.
I have one major and several minor comments:
Major:
1/ Although the authors declared that Q-flow sequence has been the standard evaluation method in these patients, the study cohort consists of only 12 patients. Therefore, the sample size is very small, the statistical power of the results maybe be influenced by it. I would recommend adding more patients into the study before submission or presenting the results somehow like “preliminary data”.
Minor:
1/ I would suggest use better the term “2D phase-contrast” then “QFlow” as it is more used and not linked with Philips scanners.
2/ The information about VENC setting is missing. Did authors use VENC in default setting or did they change it according to the individual velocities?
3/ I know that measured parameters of Qflow are named for the heart/valve analysis. I am not sure of using term “stroke volume” for the superficial venous system.
4/ I missed detailed discussion about using 2D phase-contrast sequence for the peripheral vessels analysis.
Author Response
I read with great interest the paper. The study aimed to determine if the effects of superficial venous intervention on hemo-dynamics could be quantified using QFlow sequence of MRI. They confirm on 12 patients’ differences in measured parameters between patients and healthy controls as well as in patients before and after interventions.
The topic is very interesting, the use of (primarily) cardiac sequence for venous system of lower extremities rather innovative. Using this approach offers new insight into venous system during interventions. The paper is clearly written and the results are properly discussed. The paper confirms that CMR can provide clinically relevant information.
I have one major and several minor comments:
Major:
1/ Although the authors declared that Q-flow sequence has been the standard evaluation method in these patients, the study cohort consists of only 12 patients. Therefore, the sample size is very small, the statistical power of the results maybe be influenced by it. I would recommend adding more patients into the study before submission or presenting the results somehow like “preliminary data”.
Response to major comment: Thank you for pointing out the weaknesses of this study. Although we had performed the MRI on 271 subjects in 2017- 2021, only 12 subjects completed pre-intervention MRI and also completed post-intervention MRI (see Figure 1). The data from these 12 patients are small in sample size but invaluable. We changed the title to “Venous Segmental Flow Changes After Superficial Venous Intervention Demonstrating by Quantitative Phase-Contrast Magnetic Resonance Analysis: Preliminary Data from a Longitudinal Cohort Study” and added further discussion in the Discussion section to disclosure this weakness of this study.
Minor:
1/ I would suggest use better the term “2D phase-contrast” then “QFlow” as it is more used and not linked with Philips scanners.
Response to minor comment 1: We had replaced the term "QFlow" with "2D PC-MRI". Besides, we also replaced the word "TRANCE MRI" with the "gated 3D-TSE-STIR MRA" to avoid linking with vendor-specific nomenclature.
2/ The information about VENC setting is missing. Did authors use VENC in default setting or did they change it according to the individual velocities?
Response to minor comment 2: We had added the detailed parameters of 2D PC-MRI scanning, including VENC setting (velocity encoding = 80cm/s). Please see the Materials and Methods section.
3/ I know that measured parameters of Qflow are named for the heart/valve analysis. I am not sure of using term “stroke volume” for the superficial venous system.
Response to minor comment 3: Hemodynamic analysis using 2D PC-MRI scanning was performed on axial planes of the venous segments, including the inferior vena cava, external iliac veins, femoral vein, popliteal vein, and great saphenous veins. By drawing the region of interest (ROI) on the vascular lumens (covering the whole lumen), the hemodynamic parameters will be generated by measuring the phase-shifting information of voxels within the ROI. Thus, when using quantitative 2D PC-MRI analysis for evaluating the superficial venous system, stroke volume reflects the net volume of blood that passes through the contour of ROI during one heartbeat. We added the detailed MRI scanning protocol and explained the definitions of the measured parameters of 2D PC-MRI. Please see the Materials and Methods section.
4/ I missed detailed discussion about using 2D phase-contrast sequence for the peripheral vessels analysis.
Response to minor comment 4: Reports about the application of 2D PC-MRI for peripheral vessel analysis are still rare and lack to be comprehended. In this study, hemodynamic analysis using 2D PC-MRI scanning was performed on axial planes of the venous segments. By drawing the region of interest (ROI) on the vascular lumens (covering the whole lumen), the hemodynamic parameters will be generated by measuring the phase-shifting information of voxels within the ROI. When using quantitative 2D PC-MRI analysis for evaluating the superficial venous system, stroke volume reflects the net volume of blood that passes through the contour of ROI during one heartbeat. As well as that, forward flow volume reflects the volume of blood that passes through the contour of ROI only in the positive direction (toward the head direction) during one heartbeat. We added description in the Discussion section.
Thank you for the informative and careful reviewing our manuscript. We learn lot during the revision this article.
Round 2
Reviewer 2 Report
Thank you for addressing my comments. I've no further ones.